**RESEARCH**

# Effective methods for bulk RNA-seq deconvolution using scnRNA-seq transcriptomes

Francisco Avila Cobos[1†], Mohammad Javad Najaf Panah[2†], Jessica Epps[2], Xiaochen Long[2,3], Tsz-Kwong Man[2], Hua-Sheng Chiu[2], Elad Chomsky[4], Evgeny Kiner[4], Michael J. Krueger[2], Diego di Bernardo[5], Luis Voloch[4], Jan Molenaar[6], Sander R. van Hooff[6], Frank Westermann[7], Selina Jansky[7], Michele L. Redell[2], Pieter Mestdagh[1*] and Pavel Sumazin[2*]

†Francisco Avila Cobos and Mohammad Javad Najaf Panah contributions equaly to this work.

*Correspondence:
pieter.mestdagh@ugent.be;
sumazin@bcm.edu

[1] Department of Biomolecular Medicine, Ghent University, Ghent, Belgium; Cancer Research Institute Ghent, Ghent, Belgium
[2] Department of Pediatrics, Baylor College of Medicine, Texas Children's Hospital Cancer Center, Houston, TX, USA
[3] Department of Statistics, Rice University, Houston, TX 77251, USA
[4] ImmunAi, New York, NY, USA
[5] Department Chemical, Materials and Industrial Engineering, Telethon Institute of Genetics and Medicine, University of Naples "Federico II", Via Campi Flegrei 34, 80078 Naples, Pozzuoli, Italy
[6] Princess Maxima Center for Pediatric Oncology, Utrecht, The Netherlands
[7] German Cancer Research Center, DKFZ, Heidelberg, Germany

## Abstract

**Background:** RNA profiling technologies at single-cell resolutions, including single-cell and single-nuclei RNA sequencing (scRNA-seq and snRNA-seq, scnRNA-seq for short), can help characterize the composition of tissues and reveal cells that influence key functions in both healthy and disease tissues. However, the use of these technologies is operationally challenging because of high costs and stringent sample-collection requirements. Computational deconvolution methods that infer the composition of bulk-profiled samples using scnRNA-seq-characterized cell types can broaden scnRNA-seq applications, but their effectiveness remains controversial.

**Results:** We produced the first systematic evaluation of deconvolution methods on datasets with either known or scnRNA-seq-estimated compositions. Our analyses revealed biases that are common to scnRNA-seq 10X Genomics assays and illustrated the importance of accurate and properly controlled data preprocessing and method selection and optimization. Moreover, our results suggested that concurrent RNA-seq and scnRNA-seq profiles can help improve the accuracy of both scnRNA-seq preprocessing and the deconvolution methods that employ them. Indeed, our proposed method, Single-cell RNA Quantity Informed Deconvolution (SQUID), which combines RNA-seq transformation and dampened weighted least-squares deconvolution approaches, consistently outperformed other methods in predicting the composition of cell mixtures and tissue samples.

**Conclusions:** We showed that analysis of concurrent RNA-seq and scnRNA-seq profiles with SQUID can produce accurate cell-type abundance estimates and that this accuracy improvement was necessary for identifying outcomes-predictive cancer cell subclones in pediatric acute myeloid leukemia and neuroblastoma datasets. These results suggest that deconvolution accuracy improvements are vital to enabling its applications in the life sciences.

## Background

Single-cell and single-nuclei RNA-sequencing (scnRNA-seq) technologies have revolutionized our ability to quantify cell types and cell states in healthy and disease tissues. scnRNA-seq technologies generate cell-type specific transcriptomes, with individual cells labeled, enumerated, and molecularly characterized. This, in turn, allows for comparing the cell composition of tissues and for associating changes in tissue cell-type abundances and both their molecular and clinical parameters. Examples include scnRNA-seq assays that helped identify programs for tissue development [1] and regeneration [2] and associated patient outcomes with tumor subclones [3]. scnRNA-seq assays helped reveal immune-cell-type composition differences that may dictate responses to immune checkpoint inhibition therapies [4], identified tumor subclones that acquired drug resistance during treatments [5], and identified cancer cells that adapted to evade targeted therapies [6]. scnRNA-seq assays are increasingly enabling research to identify therapeutic targets and diagnostic biomarkers in efforts to improve therapies for cancer and other diseases.

While scnRNA-seq assays can provide cell-type-specific information at unprecedented resolutions, their implementation is associated with challenges that prevent their widespread adoption in clinical settings. These challenges include the high cost of library preparation and sequencing and the stringent requirements for sample collection, processing, and storage. Namely, the current cost of scnRNA-seq assays is 10–30-fold greater than the cost of bulk RNA sequencing (RNA-seq), which effectively prevents their adoption at scales previously seen for RNA-seq. Importantly, specialized facilities for sample collection and tissue processing are required for accurate profiling, and these are not readily available at most hospitals or academic institutions. For example, accurate scRNA-seq profiles require fresh tissue dissociation and cell suspension generation at carefully controlled temperatures. Moreover, tissue preservation and cell sorting are known to alter scnRNA-seq estimates, with some commonly used methods shown to introduce bias by selectively depleting genes and cell types [7–9].

RNA-seq is less challenging to implement in clinical settings, but it only provides mean gene expression abundance estimates across cell types. Recently, computational deconvolution methods were proposed to infer cell-type abundances from RNA-seq profiles using either reference matrices composed of cell-type-specific gene expression signatures [10–12] or scnRNA-seq data from the same tissue type [13–16]. In various benchmarking efforts, we and others have shown that multiple factors, including data transformation, data normalization, and the composition of the reference matrix can impact the performance of deconvolution methods [10, 17]. However, given the potential impact of scnRNA-seq-based deconvolution on advances in the life sciences, there remains a need to systematically compare and quantify the absolute accuracies of deconvolution methods.

Here, we evaluated deconvolution methods in 8 datasets of concurrent bulk RNA-seq and scnRNA-seq profiles (see Additional file 5: Table S4). These datasets included cell mixtures, where cell type abundances and expression profiles are known with high accuracy and that could be used to quantify both deconvolution and scRNA-seq expression estimates, as well as tissues that allow comparing the effects of common preservation protocols. When evaluating deconvolutions of bulk RNA-seq profiles, accuracy

was determined by comparing deconvolution-predicted cell abundances to gold-standard estimates, where gold-standard estimates were derived from either validated counts of the composing cells or the analyses of scnRNA-seq profiles. Surprisingly, our results suggested that some methods consistently produced the most accurate cell-abundance estimates, irrespective of datasets or data processing.

We hypothesized that concurrent RNA-seq and scnRNA-seq profiling could be used to not only evaluate deconvolution methods but also improve deconvolution accuracy. To test this, we developed the R package Single-cell RNA Quantity Informed Deconvolution (SQUID), which combines bulk RNA-seq transformation and dampened weighted least squares deconvolution approaches. Analyses of SQUID accuracy suggested that methods that harness the power of concurrent RNA-seq and scnRNA-seq profiling can consistently outperform other methods in predicting the composition of cell mixtures and tissue samples. Finally, to evaluate the benefit of improved deconvolution accuracy for applications in cancer research, we concurrently profiled pediatric acute myeloid leukemia (AML) and neuroblastoma samples by RNA-seq and scRNA-seq and tested whether deconvolution methods can predict risk based on the abundance of potential high-risk cancer subclones in diagnostic samples. Our results indicated that only SQUID subclone-abundance estimates were predictive of outcomes in RNA-seq-profiled AML and neuroblastoma diagnostic samples. Thus, we concluded that SQUID's deconvolution-accuracy improvement is key to enabling its potential applications in diagnostic protocols for these cancers.

## Results

To quantify absolute deconvolution performance, we established a framework based on concurrent bulk RNA-seq and scRNA-seq or snRNA-seq data across human and murine tissues. In parallel, we evaluated the impact of RNA-seq and scnRNA-seq data normalization strategies on deconvolution performance (Fig. 1). While concurrent RNA-seq and scnRNA-seq assays can be used to evaluate deconvolution accuracy, they lack controls for both true composition and cell-type expression estimates. Namely, divergent estimates from the two assays cannot be resolved, and technical analysis errors may not be identified due to missing information. Consequently, accurate and fully resolved deconvolution-strategy evaluations require fully characterized datasets, where the expression profiles and composition of each cell type are known with high degrees of accuracy. To accomplish this, we developed a solid tumor model that includes multiple solid-tumor cell types, immune cells, and lower-abundance stem cells. We then generated and concurrently profiled cell mixtures that conform to this model by flow cytometry, RNA-seq, and scRNA-seq. Here, we present the results of our efforts to evaluate deconvolution methods on cell mixtures and tissue samples and evaluate whether improved deconvolution accuracy can benefit its potential applications in diagnosing cancer patients.

### Cell mixtures characterization

We established six in vitro cell mixtures that are composed of varying proportions of cells from 3 breast cancer lines (T47D, BT474, MCF7), monocytes (Thp1), lymphocytes (Jurkat), and stem cells (hMSC). Mixture composition was recorded based on input cell counts. Cells from each cell line and cells from each mixture were profiled by bulk

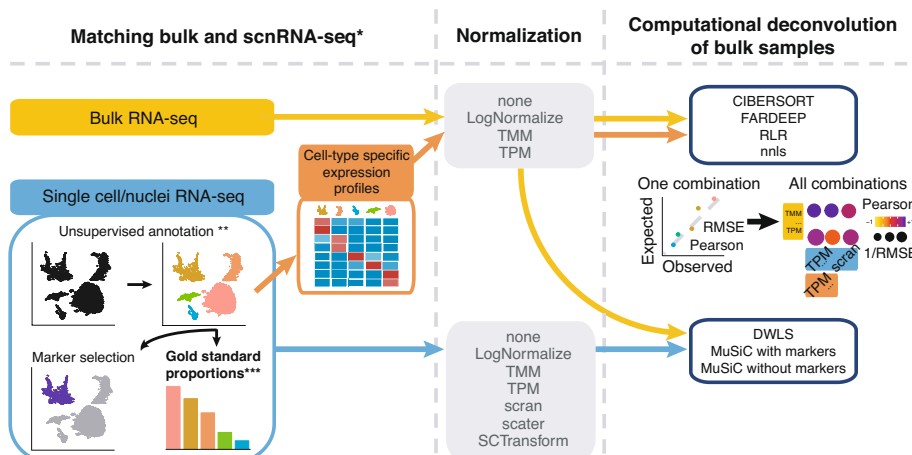

**Fig. 1** We benchmarked data normalizations and deconvolution approaches in datasets with concurrent bulk RNA-seq and scnRNA-seq profiles (*). Cells were clustered in an unsupervised fashion (**). Gold standard abundance estimates (***) for each cell type were obtained by either aggregating cells or nuclei in each scnRNA-seq cluster, immunohistochemistry, fluorescence-activated cell sorting, or cell counts. Deconvolution methods used either full scnRNA-seq expression profiles or cluster-specific biomarkers to predict cell-type abundances based on bulk RNA-seq profiles. Deconvolution accuracies in each sample were assessed by comparing predicted abundances from bulk RNA-seq and gold standard estimates

RNA-seq in triplicates. Cell mixtures were profiled by flow cytometry in triplicates to independently evaluate their composition (Fig. 2A, Supplemental Table 1). The proportions of breast cancer cell lines varied across mixtures, with some mixtures composed of predominantly one cell type (e.g., 66% of mixture 1 were T47D cells) and others having a balanced composition (e.g., mixture 4). Monocytes and lymphocytes accounted for 15% of the mixtures, and hMSC abundances varied from 0.5% to 2% (Fig. 2B, Additional file 2: Table S1). See Methods for detailed experimental descriptions.

UMAP analysis of mixture scRNA-seq profiles verified the existence of 6 clusters with biomarkers that correspond to their six composing cell types (Fig. 2C, Additional file 3: Table S2). We confirmed breast-cancer cell type identities by integrating seven scRNA-seq profiles of breast-cancer cell samples [18], including T47D, BT474, MCF7, and four cell lines that were not used in our mixtures (BT483, AU565, HCC70, DU4475); see Fig. 2D. Cellular composition estimates based on absolute cell counts that were determined during mixture assembly showed high correlations with composition estimates by flow cytometry and scRNA-seq: $r = 0.97$ and $r = 0.96$, respectively, Fig. 2E and F. However, the correlation between estimates by flow cytometry and scRNA-seq clusters was significantly lower ($r = 0.92$, $p < 0.05$ by Fisher's transformation). This suggested that composition estimates by cell counts are the most accurate, and flow cytometry and scRNA-seq introduce independent errors to composition estimates. Overall, however, these results confirmed the mixture composition as estimated by cell counting and demonstrated that it is reflected in scRNA-seq data with good accuracy. Note that the accuracy of scRNA-seq-derived expression estimates of individual cell types was not as good as the corresponding mixture composition estimates. Specifically, Pearson correlation of the profiles of the predicted T47D, BT474, and MCF7 cells and their respective bulk RNA-seq profiles were $r = 0.53$, $r = 0.53$, and $r = 0.55$, respectively; Jurkat and Thp1 had Pearson correlations

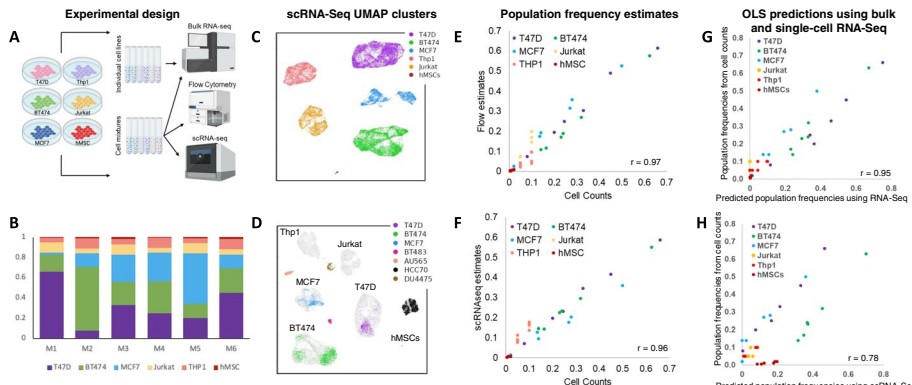

**Fig. 2** Cell mixture design, characterization, and analysis by OLS. **A** Breast cancer cells (BT474, T47D, and MCF7), leukemia cells (THP1 and Jurkat), and human mesenchymal stem cells (hMSCs) were used to generate six populations of mixed cells (cell mixtures). Each cell line was profiled individually by bulk RNA-seq in triplicates, and each mixture was profiled by bulk RNA-seq and flow cytometry in triplicates as well as by a 10 × genomics Chromium controller. **B** Cell mixtures were composed of varying proportions of cancer cells, with leukemia cells accounting for 15% and hMSC accounting for 0.5% (M1 and M4) to 2% (M3 and M6) of each mixture. **C** The clusters derived from scRNA-seq data corresponded to composing cell types, as identified by cell-type biomarkers. **D** The integration of scRNA-seq profiles of our mixtures (in gray) and scRNA-seq profiles of BT474, T47D, MCF7, BT483, AU565, HCC70, and DU4475 (Gambardella et al., 2022) revealed a significant overlap between profiles of BT474, T47D, and MCF7 cells, while negative controls, including BT483, AU565, HCC70, and DU4475, clustered separately. **E** Cell counts at the time of mixture generation were significantly correlated with cellular composition estimates by flow cytometry ($r = 0.97$) and **F** by scRNA-seq analysis ($r = 0.96$). However, the correlation between the estimates by flow cytometry and scRNA-seq was significantly lower ($r = 0.92$, $p < 0.05$, Fisher's transformation). **G** Ordinary least squares regression (OLS) using bulk RNA-seq profiles of composing cell types estimated the composition of our mixtures with high accuracy ($r = 0.95$). **H** OLS deconvolution abundance estimates using cell-type expression profiles from scRNA-seq analysis were also accurate ($r = 0.72$, $p < 1E - 4$) but significantly worse ($p < 1E - 5$, Fisher's transformation)

of $r = 0.66$ and $r = 0.63$, respectively; hMSCs, which were the least abundant cells in each mixture, were correlated at $r = 0.16$ with their bulk profiles. Moreover, restricting comparisons to the top expressed genes did not improve these correlations (Additional file 4: Table S3).

### Cell mixtures reveal differences in deconvolution accuracy

To evaluate the effects of expression-estimate inaccuracies on the quality of deconvolution, we tested the accuracy of ordinary least squares regression (OLS) in predicting mixture composition from its bulk profiles and using either scRNA-seq or bulk-derived expression profile estimates for each cell type. Our results suggested that OLS can estimate mixture composition with high accuracy when input expression profile estimates are accurate. Namely, using bulk RNA-seq profiles of each cell type, OLS composition predictions had Pearson correlations of $r = 0.95$ with mixture composition estimates by cell counts (Fig. 2G). However, when using scRNA-seq-based expression estimates for each cell type, this correlation declined to $r = 0.78$ (Fig. 2H). Note that the correlation $r = 0.78$ is significant at $p < 1E - 5$, suggesting that, overall, OLS can predict composition in our mixtures using scRNA-seq-based expression estimates. However, its deconvolution accuracy using scRNA-seq-based expression estimates was significantly lower than

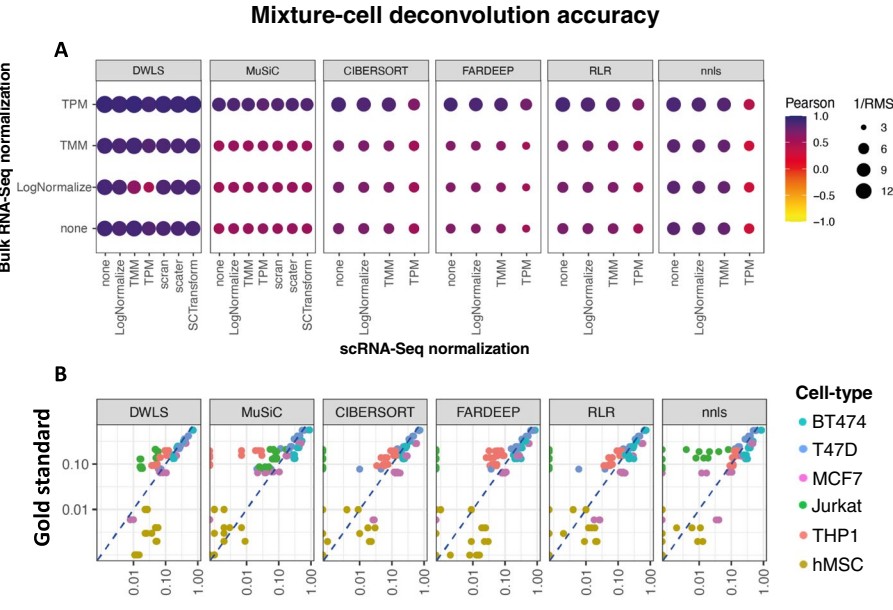

**Fig. 3** The accuracy of cell-mixture deconvolution. **A** The impact of RNA-seq and scRNA-seq normalization strategies and the choice of deconvolution methods on deconvolution accuracy, as assessed by Pearson correlation and root mean square error (RMSE); darker and larger circles represent higher Pearson and lower RMSE values, respectively. **B** Deconvolution results for the normalization strategy with the lowest RMSE; axes are in $\log_{10}$ scales. Each scatterplot contains 36 data points corresponding to 6 cell lines in 6 mixtures, with gold standard abundance estimates based on cell counts and predicted abundances based on deconvolution

when using bulk RNA-seq-based expression estimates. As expected, hMSC composition estimates were the least accurate (Fig. 2H).

Having confirmed the quality and validity of the in vitro cell mixtures and associated data, we used our benchmarking framework to evaluate deconvolution methods on these data (Fig. 3A). We observed substantial differences in performance (i.e., predicted abundances versus gold standard) between methods, with dampened weighted least squares (DWLS) outperforming the other five methods, irrespective of the bulk RNA-seq and scRNA-seq normalization strategy. Overall, normalization of the bulk RNA-seq data with TPM resulted in better performance compared to TMM, LogNormalize, or when no normalization was applied. Normalization of the scRNA-seq-derived reference matrix had a lower impact on deconvolution accuracy. All methods performed poorly in predicting the abundance of hMSC cells (Fig. 3B). All methods also underestimated the fraction of Jurkat cells in several mixtures, but this was most pronounced for CIBERSORT and OLS with a non-negativity constraint (NNLS). In addition, MuSiC underestimated the fraction of THP1 cells. Together, these observations demonstrated that, in an ideal setting, with concordant scRNA-seq and bulk RNA-seq, deconvolution with DWLS leads to the most accurate cell-type abundance estimates.

### Variable deconvolution accuracy across human tissues

We studied seven human and murine tissue datasets with concurrent RNA-seq and scn-RNA-seq profiles. DWLS outperformed the other methods in 6/7 datasets with higher

Pearson correlation coefficients and lower root mean squared error (RMSE, Fig. 4A and B). The absolute performance of all methods was very high in the remaining dataset (Brain, see Fig. 4A and B). Despite DWLS outperforming the other methods, its absolute performance differed substantially across datasets. DWLS performance was high in the fresh kidney, AML, NB1, and brain datasets but lower in the NB2, breast cancer, and synapse datasets, with average Pearson correlation coefficients above 0.67 and below 0.4, respectively. The choice of data normalization method impacted deconvolution performance in a subset of datasets, but the impact on performance was typically modest, and none of the normalization methods consistently performed better or worse across datasets.

### Single-cell storage procedures impact deconvolution accuracy

Procedures to store single-cell suspensions are known to alter cell type abundance estimates by scRNA-seq [7]. To evaluate the impact of cell storage procedures on deconvolution accuracy, we compared deconvolution performance on two datasets—mouse kidney and human breast cancer—with concurrent bulk profiles and technical replicate scRNA-seq profiles of single-cell suspensions derived from alternative tissue preservation methods. The kidney dataset included scRNA-seq profiles of methanol-fixed, cryopreserved, and fresh tissues [19], and the breast cancer dataset included profiles of fresh and cryopreserved tissues [20, 21]. We applied deconvolution on the matching bulk RNA-seq data using DWLS and FARDEEP—these methods performed relatively

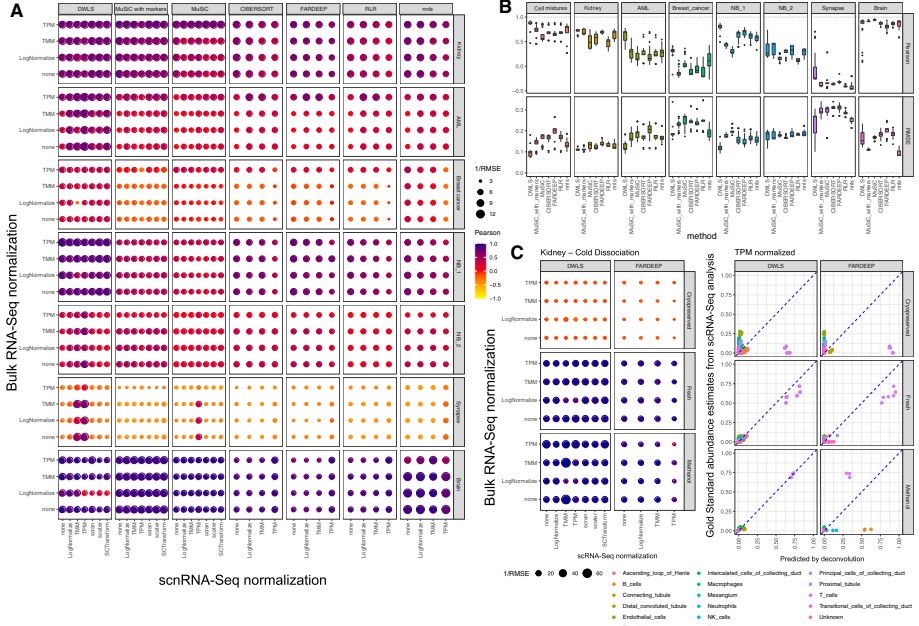

**Fig. 4** Deconvolution accuracy on concurrently profiled tissues. **A** The impact of bulk RNA-seq and scRNA-seq normalization strategies on the accuracy of deconvolution methods, as assessed by Pearson correlation and root mean square error (RMSE); darker and larger circles represent stronger correlations and smaller errors, respectively. **B** Numerical visualizations of deconvolution accuracies in cell mixtures and tissues across normalization strategies, as assessed by Pearson correlation (top) and RMSE (bottom). **C** The effects of profiling cryopreserved, fresh, and methanol-preserved cold-dissociated kidney samples on the accuracies of deconvolution by DWLS and FARDEEP

well in our tests—and compared predicted cell type abundances to the gold standard in each of the scRNA-seq datasets. Both DWLS and FARDEEP showed good performance when comparing observed cell type abundances to those in the fresh and methanol-fixed kidney tissues, but both performed poorly when compared to the gold standard for the cryopreserved scRNA-seq dataset (Fig. 4C). We note that while the overall deconvolution accuracy for the breast cancer dataset was lower than that of the kidney dataset, there remained a significant difference in performance between fresh suspensions and cryopreserved suspensions (Additional file 1: Figure S2). Because of the variability in deconvolution accuracy, we included a comparison of all deconvolution and normalization methods for the kidney dataset (Fig. 4C and Figure S5). We note that while the normalization choice had a relatively small impact on deconvolution accuracy for the kidney dataset, normalization may have had a strong effect on the performance of each tested method for the breast cancer dataset (Fig. 4A and Figure S2). However, our accuracy estimates were dominated by few abundant cell lines and may not generalize.

### Transformation of bulk RNA-seq data with SQUID improves deconvolution accuracy

DWLS consistently outperformed other deconvolution methods in our tests. However, its accuracy was poor in several datasets, limiting its potential applications. Note that lower accuracy may be due to method-independent factors, including physically different cellular compositions between scnRNA-seq and bulk RNA-seq samples, and technical differences in sample processing that results in diverging estimates. Most importantly, deconvolution accuracy is dependent on accurate gene expression estimates, and—as is the case for our cell mixtures—scnRNA-seq-derived gene expression profiles may be imprecise. Indeed, we showed that OLS-based deconvolution using bulk RNA-seq profiles of each cell type (Fig. 2G) produced more accurate results than deconvolution using scRNA-seq-derived profiles (Fig. 2H) on our cell mixtures. Similarly, deconvolution with DWLS using bulk RNA-seq profiles of each cell type was in excellent agreement with mixture composition as estimated by cell counts (Fig. 5A), and its performance declined when using scRNA-seq-derived profiles (Fig. 5B). We note that the same was observed when estimating mixture abundances using either scRNA-seq analysis or flow cytometry. However, in all cases, deconvolution with DWLS was more accurate than OLS. Relative to cell-count estimated mixture abundances, DWLS and OLS predictions had $r=0.98$ and RMSE$=0.04$ vs. $r=0.95$ and RMSE$=0.06$ when using bulk RNA-seq profiles, and $r=0.93$ and RMSE$=0.08$ vs. $r=0.78$ and RMSE$=0.12$ when using scRNA-seq-derived profiles, respectively. Based on these observations, we attempted to further improve DWLS performance by transforming bulk RNA-seq profiles to scRNA-seq vector spaces. This approach, which we coined "SQUID," employed linear bulk RNA-seq transformation followed by dampened weighted least squares, and it further improved deconvolution accuracy ($r=0.95$ and RMSE$=0.06$, Fig. 5C).

To systematically test the benefit of bulk transformation and deconvolution with SQUID, we compared the performance of SQUID, DWLS, and OLS for our cell mixtures, as well as for pediatric AML, NB1, NB2, Synapse (ROSMAP brain), breast cancer, and kidney datasets using a leave-one-out cross-validation strategy. Namely, iteratively, concurrent RNA-seq and scnRNA-seq profiles of all but one of the samples were used to predict the composition of the remaining sample based on its bulk RNA-seq profile

**Fig. 5** Mixture deconvolution with transformed RNA-seq data. **A** DWLS deconvolved the composition of our mixtures with near-perfect accuracy when given the bulk RNA-seq expression profiles of each cell type ($r = 0.98$, RMSE $= 0.04$) and **B** with high accuracy when using cell-type expression estimates from scRNA-cluster profiles ($r = 0.93$, RMSE $= 0.08$). **C** SQUID deconvolution accuracy, relative to cell counts, when using cell-type expression estimates from scRNA-cluster profiles ($r = 0.95$, RMSE $= 0.06$) was significantly better than DWLS ($p < 2E - 4$, Fisher's transformation). **D** Deconvolution accuracies of concurrent RNA-seq and scnRNA-seq profiled tissues using SQUID, DWLS, and OLS, as assessed by Pearson correlation and root mean square error (RMSE)

(Fig. 5D). Our results suggested consistently and significantly improved prediction accuracies with SQUID. Comparisons of SQUID accuracy with the other methods, including DWLS, CIBERSORT, FARDEEP, RLR, NNLS, MuSiC, and Bisque (Additional file 1: Figure S6) without cross validation—analogous to Fig. 4 comparisons—are given in Additional file 1: Figures S2-6.

### Deconvolution of pediatric AML and neuroblastoma dataset with SQUID

To assess the utility of deconvolution on bulk RNA-seq of clinical samples, we focused on profiles of pediatric AML and neuroblastoma samples. Large-scale clinical and bulk

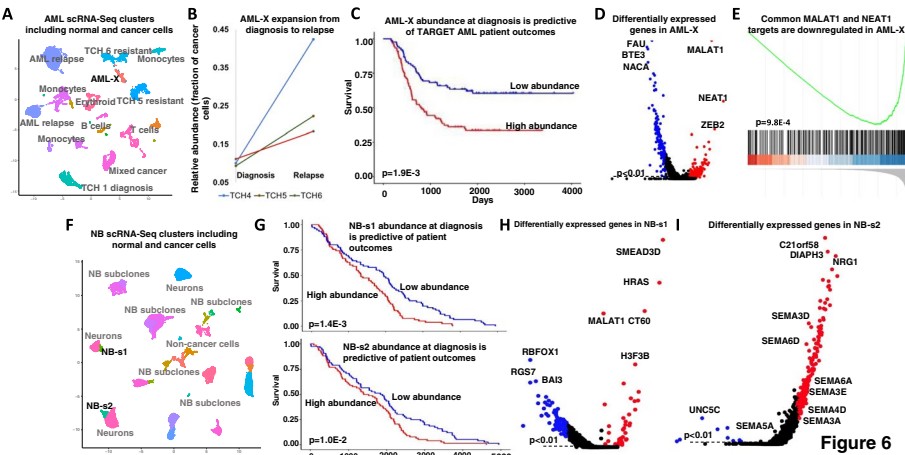

**Fig. 6** SQUID-predicted cell-type abundances identified outcomes-predictive subclones in pediatric AML and neuroblastoma diagnostic biopsies. **A** Clusters identified in scRNA-seq profiles of paired diagnostic and relapse samples from 6 pediatric AML patients included the cluster AML-X. **B** AML-X cells were present in the diagnostic biopsies of 3 patients and their abundance increased at relapse. **C** SQUID-predicted AML-X abundances in TARGET-profiled diagnostic AML biopsies were predictive of patient outcomes. **D** The cancer lncRNAs MALAT1 and NEAT1 and **E** their direct targets were upregulated in AML-X and were predicted to regulate chemoresistance in AML. **F** Clusters identified in scRNA-seq profiles of neuroblastoma samples included NB-s1 and NB-s2. **G** SQUID-predicted abundances of both NB-s1 and NB-s2 in TARGET-profiled diagnostic biopsies were predictive of patient outcomes. **H** Upregulated genes in NB-s1 included SEMA3D and HRAS, and upregulated genes in NB-s2 (**I**) included SEMA3D and other semaphorin family members

RNA-seq profiles are available for both these tumor types from the TARGET consortium, including the profiles of 181 pediatric AML [22] and 161 neuroblastoma [23] patient samples. We profiled paired diagnostic pre-treatment and relapse samples for 6 AML patients using concurrent RNA-seq and scRNA-seq assays, and we profiled the expression of 14 neuroblastoma samples using bulk RNA-seq; note that we previously reported on the scRNA-seq profiles of the 14 neuroblastomas [24] and used it here to evaluate deconvolution accuracy (the NB1 dataset). Pre-treatment AML samples were expected to be enriched for chemosensitive cancer cells, while relapse AML samples were expected to be enriched for chemoresistant cancer cells [25]. We used predicted cell types and expression profiles from these scRNA-seq data to deconvolve RNA-seq profiles of TARGET AML and neuroblastoma diagnostic samples.

Paired diagnostic-relapse pediatric AML samples were collected to identify chemoresistant tumor subclones. After integration and clustering (Fig. 6A), we sought to identify AML subclones (clusters) that are present before treatment and expand at relapse. We found one AML cluster that included diagnostic and relapse cells from at least half of the patients and expanded at relapse. We refer to this subclone as the AML expanding subclone, or AML-X for short (Fig. 6B). We used SQUID, DWLS, CIBERSORTx, and OLS to predict the composition of TARGET AML samples from chemotherapy trials AAML03P1 (40 patients), AAML0531 (171 patients), and CCG-2961 (24 patients). We then used the predicted abundance of AML-X cells in each diagnostic sample to predict patient outcomes by survival analysis. Note that AAML03P1 and AAML0531 patients were treated with a variety of chemotherapy and CD33-inhibitor combinations, and the earlier CCG-2961

patients were treated by combinations of chemotherapy and anthracyclines. Abundance estimates by DWLS, CIBERSORTx, and OLS were not predictive of outcomes. However, cell-type abundance estimates by SQUID suggested that diagnostic samples whose composition included at least 5% AML-X cells had significantly worse outcomes ($p = 1.90E-3$, Kaplan–Meier estimator, Fig. 6C). SQUID composition estimates were also the only ones that were predictive of survival by Cox regression ($p = 6E-4$, compared to $p = 0.07$, $p = 0.41$, and $p = 0.68$ using DWLS, CIBERSORTx, and OLS, respectively). Note AML-X cells accounted for ~ 10% of the AML cells in the three scRNA-seq-profiled diagnostic samples with AML-X cells; this composition estimate was consistent with SQUID estimates in TARGET diagnostic samples after accounting for tumor purity (Additional file 6: Table S5). The most upregulated genes in AML-X were MALAT1, NEAT1, and ZEB2 (Fig. 6D). The long non-coding RNAs NEAT1 and MALAT1 co-localize in Chr11Q13.1, are co-expressed in pediatric AML, and are predicted to transcriptionally co-inhibit hundreds of genes [26, 27]. Their common targets were significantly downregulated in AML-X (Fig. 6E). Moreover, NEAT1 has been previously implicated with chemoresistance in cancer [28], and both NEAT1 and MALAT1 have been associated with poor prognosis in childhood leukemia [29]. In addition, MALAT1 has been shown to post-transcriptionally upregulate ZEB2 in cancer [30, 31], and ZEB2 was the third most upregulated gene in AML-X.

To identify neuroblastoma cell clusters that are associated with outcomes, we integrated scRNA-seq data across the 14 neuroblastoma samples from the NB1 dataset and identified 15 cell clusters (Fig. 6F). Each cluster was tested for patient outcomes prediction based on abundance estimates by SQUID, DWLS, CIBERSORTx, and OLS using TARGET RNA-seq data (Additional file 6: Table S5). The target dataset includes profiles of 161 samples from 69 clinical trials where patients were treated by combinations of a variety of chemotherapies and other therapies, including GD2 and thymidylate-synthase inhibitors. In total, two cell clusters were identified to be predictive of outcomes using SQUID abundance estimates (Fig. 6G, Cluster NB-s1 at $p = 1.4E-3$ and Cluster NB-s2 at $p = 1.0E-2$, Kaplan–Meier estimator). No cluster was predictive of outcomes using estimates from other survival methods: DWLS, CIBERSORTx, and OLS abundance estimates for NB-s1 were predictive of survival at $p = 0.23$, $p = 0.95$, and $p = 0.99$, and for NB-s2 at $p = 0.34$, 0.93 and $p = 0.99$, respectively. Notably, among the top 5 most upregulated genes in cluster NB-s1 were HRAS, SEMA3D, and H3F3B (Fig. 6H). RAS pathway mutations have previously been identified in relapsed neuroblastomas [32]. More recently, upregulation of H3F3B has been associated with the alternative lengthening of telomeres (ALT) phenotype in neuroblastoma, which is associated with poor outcomes [33]. Moreover, tumors harboring RAS pathway mutations in combination with telomere maintenance mechanisms were shown to have extremely poor survival rates [34]. In cluster NB-s2, we observed the upregulation of 6 members of the semaphorin family, including SEMA3D (Fig. 6I). SEMA3D upregulation has been documented in metastatic neuroblastomas and was shown to affect neuroblastoma cell migration [35].

## Discussion

Profiling technologies at single-cell resolutions are enabling efforts to characterize the cellular composition of complex tissues. Single-cell resolution RNA profiling technologies, including 10X Genomics platforms, are used to characterize the transcriptomes

of individual cells, which, in turn, can be used to identify these cell types in past and future assays. Consequently, ongoing large-scale efforts, including The Human Cell Atlas [36], single-cell tumor immune atlas [37], and single-cell Atlas in Liver Cancer [38], are mapping out healthy and disease tissues and characterizing the transcriptomes of their composing cell types. These efforts are building resources that promise to improve our understanding of intercellular dependencies between healthy and diseased cells and to enable comparisons of tissues at high resolutions. Single-cell atlases promise to help interpret future single-cell assays and help maximize knowledge gained from RNA-seq profiles. RNA-seq profiles remain by far the most frequently used type of molecular data collected in the biological and health sciences, and they account for more publicly available molecular datasets than any other data type. Because of the technical and financial challenges associated with scnRNA-seq, RNA-seq is likely to remain the most frequently used assay for the foreseeable future. Consequently, computational deconvolution of bulk transcriptomes could serve as an alternative for scnRNA-seq to enumerate cell types in complex tissues, including tumor biopsies. We note that while we used the scnRNA-seq abbreviation to improve readability, scRNA-seq and snRNA-seq assays can produce substantially different expression estimates because the RNA populations profiled by these assays are not the same. Our study was not sufficiently powered to evaluate differences between deconvolutions based on scRNA-seq and snRNA-seq assays, but we expect that their expression estimates are sufficiently correlated to allow for accurate and consistent cell-type mappings between scRNA-seq and snRNA-seq profiles.

Deconvolution methods that use scnRNA-seq profiles to predict the composition of bulk-profiled samples are expected to play major roles in analyses based on single-cell atlases. However, their absolute accuracy remains understudied, and their potential users face multiple unaddressed challenges. First and foremost, current deconvolution methods are heuristics that always produce composition estimates irrespective of accuracy. Most methods do not provide accuracy evaluations, and efforts to evaluate accuracy will have limited success without assay-specific quality controls, which are not always available. Other challenges include the lack of guidance for choosing technical parameters in data analysis, including the choice of methods and parameters for data normalization, data harmonization, and clustering. These choices dictate the accuracy of scnRNA-seq analysis and its use for deconvolution. In summary, the deconvolution of RNA-seq profiles based on scnRNA-seq data will benefit from reliable accuracy evaluation and guidance for selecting analysis parameters and methods.

Here, we produced comparative performance analyses of deconvolution methods based on constructed cell mixtures with known cell abundances and expression profiles and based on concurrent scnRNA-seq and bulk RNA-seq data across a variety of tissue types. Our analyses of cell-mixtures samples suggested that current scRNA-seq assays using the $10 \times$ Genomics platform can produce excellent sample-composition estimates, but these assays may produce relatively poor transcriptome characterizations for each identified cell type and particularly for rare cell types. Moreover, our results suggested that scRNA-seq assays tend to under-sample adherent cells when non-adherent cells are present. We showed that when given accurate cell-type expression profiles, direct approaches like OLS for predicting cell-type abundances from bulk profiles produced excellent results (Fig. 2G). However, deconvolution using scnRNA-seq-derived profiles

using the same approach produced poor cell-type abundance estimates (Fig. 2H). Such drops in accuracy may be due to technical and biological biases, including cellular RNA content [39], but in this case, they appear to be driven by systematic cell-dependent inaccuracies in scRNA-seq gene expression estimates. We note that our cell mixture model is a simplification of solid tumor samples. Namely, biological samples are often composed of more than six cell types, the catalog of cell types composing any two samples is often not equivalent, and tumor subtypes are likely to be more closely related than cell lines that are derived from unrelated patients. Moreover, in Fig. 2, we focused on the analysis of OLS-based deconvolution, but other deconvolution methods—including MuSiC, DWLS, and SQUID—had substantially more accurate abundance estimates than OLS.

Our results suggested that accurate evaluations and performance of scnRNA-seq-based deconvolution methods for any given context will greatly benefit from the collection of concurrent scnRNA-seq and bulk RNA-seq data. Namely, bulk RNA-seq profiles allowed us to produce upper bounds on the accuracy of deconvolution methods that rely on the corresponding scnRNA-seq assays, and the integration of concurrent bulk RNA-seq in the deconvolution process with SQUID improved deconvolution accuracy for all datasets. In addition, we observed substantial and consistent performance differences that were associated with library preparation methods, as well as analysis and deconvolution methods. Namely, comparisons of related datasets—e.g., our two neuroblastoma datasets—suggested that datasets with few scnRNA-seq profiles lead to worse deconvolution accuracy. We note that while the choice of scnRNA-seq normalization methods influenced deconvolution methods performance in some datasets, the best choices varied across datasets and deconvolution methods. For example, while LogNormalize led to good performance for most deconvolution methods in our cell-mixture scRNA-seq data, it was associated with reduced DWLS accuracy (Fig. 3A). Overall, TPM normalization produced some of the most consistent results. However, the resolution of scnRNA-seq clustering had a greater influence on deconvolution success. Namely, high clustering resolutions could lead to reduced deconvolution accuracy when multiple cell clusters share the same cell type and have highly similar transcriptomes, while low cluster resolutions could lead to heterogenous cell clusters that are not associated with specific cell types. In both cases, cell-type specific deconvolution marker genes were difficult to identify and had limited cell-type selectivity. To resolve this, we opted to either merge clusters with similar transcriptomes or select resolutions to optimize the accuracy of OLS deconvolution of concurrent RNA-seq profiles. Both approaches lead to dramatic improvements in deconvolution accuracy for all methods.

We developed a deconvolution strategy with substantially improved accuracy using concurrent scnRNA-seq and bulk RNA-seq profiles. Jew et al. suggested that the transformation of bulk RNA-seq profiles to scnRNA-seq space could improve the accuracy of RNA-seq deconvolution, and their proposed method Bisque [40] combined RNA-seq transformation and NNLS to predict the composition of RNA-seq profiled samples based on scnRNA-seq profiles. However, by combining RNA-seq transformation with the dampened weighted least squares strategy employed by DWLS [14], we were able to dramatically improve deconvolution accuracy. Indeed, our proposed strategy (SQUID) outperformed all other strategies on all datasets with or without cross validation (Fig. 5 and Additional file 1: Figures S2-6, respectively), and when estimating cell abundances

based on IHC, cell counts, flow cytometry, or scnRNA-seq analyses. We note that the number of publicly available deconvolution methods is increasing rapidly, and not all methods were evaluated here. We selected methods based on their demonstrated accuracy in previous studies and based on our ability to integrate them into our benchmark implementation. Consequently, because the Jew et al. (2020) implementation of Bisque expects raw counts from scnRNA-seq profiles, we could not include it in most of our evaluation, and it only appears in comparisons presented in Additional file 1: Figure S6.

To investigate the effects of tissue preservation methods on deconvolution accuracy, we evaluated deconvolution methods using scRNA-seq profiles of matched suspensions derived from methanol fixed, cryopreserved, or fresh tissues. We showed that deconvolution based on scRNA-seq profiles of fresh and methanol-fixed tissues can perform with good accuracy, but performance based on matched cryopreserved samples was markedly worse. These results are in line with observations made by Denisenko et al. that cryopreservation resulted in the loss of proximal tubule (epithelial) cell types, while methanol fixation maintained cellular composition [7]. Consequently, cryopreservation distorted abundance estimates, leading to a poor correlation between the predicted and expected cell type abundances. Interestingly, while not as accurate as using fresh or methanol-fixed profiles, SQUID predictions based on profiles of cryopreserved samples were dramatically more accurate than other deconvolution methods. This was due, in part, to its employment of RNA-seq transformation, which transformed bulk profiles to mirror cell-type depletions observed in scRNA-seq profiles. Thus, while SQUID reduced the discrepancy between the concurrent profiles, it did not fully correct scRNA-seq inaccuracies. We argued that because scRNA-seq profiles can include inaccuracies, frameworks to evaluate deconvolution need to include datasets where both expression profiles and cell-type abundances are fully characterized, as in our mixture data.

Finally, we showed that improved deconvolution accuracy may be necessary for enabling its applications in cancer diagnostics. To evaluate this, we produced concurrent RNA-seq and scRNA-seq profiles for pediatric AML and neuroblastoma samples and analyzed RNA-seq profiles and clinical annotations from TARGET-profiled samples. We identified a potentially chemoresistant pediatric AML subclone by comparing scRNA-seq profiles of matching diagnosis and relapse samples, and we generated subclone characterizations for neuroblastoma. We showed that only SQUID-predicted tumor subclone abundances in diagnostic samples were predictive of patient outcomes. Interestingly, while composition estimates by other methods failed to associate subclone abundances and patient outcomes in these datasets, the significance of outcomes predictions based on abundance estimates by DWLS, CIBERSORTx, and OLS mirrored their estimated accuracy in our benchmark. Namely, following SQUID, DWLS-predicted abundance estimates for our candidate high-risk subclones were the most predictive of outcomes, while OLS estimates were the least predictive.

## Conclusions

We identified key prerequisites and provided guidance for producing accurate deconvolution of RNA-seq profiled tissues based on scnRNA-seq datasets. We found that scnRNA-seq-based composition estimates are often inaccurate for cryopreserved tissues, that expression-normalization methods should be selected in a context-specific manner,

and that cell-clustering resolutions should be carefully calibrated. Our analyses suggested that, albeit at a marginally higher cost than scnRNA-seq profiles alone, concurrent RNA-seq and scnRNA-seq profiles could be used to optimize normalization and clustering, evaluate the accuracy of deconvolution methods, and improve deconvolution accuracy. Taken together, our results suggested that RNA-seq deconvolution using scnRNA-seq data can produce accurate cell-type abundance estimates and that atlases of concurrent RNA-seq and scnRNA-seq profiles could be used to reevaluate the compositions of other RNA-seq datasets.

## Methods

### Deconvolution benchmarking framework

For those scnRNA-seq datasets for which no metadata or cell label information was available, cells were clustered together in an unsupervised fashion using Monocle3. Specifically, we sequentially applied the "preprocess_cds" (num_dim = 100, norm_method = "log," method = "PCA," scaling = TRUE), "reduce_dimension" (max_components = 2, umap.metric = "cosine", umap.fast_sgd = FALSE, preprocess_method = 'PCA') and "cluster_cells" ($k$ = 20, resolution = NULL, partition_qval = 0.05, num_iter = 1) functions; see our GitHub repository for detailed code, functions, and parameters. During quality control and preprocessing, we removed cells with extreme mitochondrial or ribosomal content (top 0.5% and bottom 0.5%), and we kept detectable genes that were expressed in at least 10 cells or 1% of the cells in any of the clusters. Next, cluster-specific gene expression profiles were obtained by averaging raw gene expression values across all cells from a given cluster, and cluster-specific markers were obtained using the FindAllMarkers function from Seurat v4.0.4 with the threshold of log2(1.5) and using a Wilcoxon test on TMM normalized scnRNA-seq data. Gold standard abundance estimates were obtained either as the sum of individual cells or nuclei present in each cluster or by immunohistochemistry/Fluorescence-activated Cell Sorting (FACSymphony) cell counts; see Fig. 1 for a schematic representation of the benchmarking framework.

We refer to cell clusters as cell types throughout the manuscript, even when no annotations are available. Cell-type specific gene signatures were used to establish reference matrices for the deconvolution of their matching bulk RNA-seq data using CIBERSORT [11, 41], FARDEEP [42], RLR [43], and NNLS [44]. Alternatively, deconvolution of bulk RNA-seq data was performed with OLS, DWLS [14], and MuSiC [15], which directly use the scnRNA-seq data as the reference. Of note, MuSiC was tested in two different ways: (1) using the markers found by FindAllMarkers described above and (2) without including any prior marker information (markers = NULL). We used OLS and NNLS as naïve deconvolution tools to benchmark all other methods. Performance was quantified by calculating the Pearson correlation coefficient and RMSE between the cell-type proportions observed by deconvolution and the expected cell-type proportions that were either known or derived from scnRNA-seq.

### Deconvolution with OLS

OLS was used to solve a simple set of linear equations that seeks to find the optimal composition *P* of a set of mixtures with bulk RNA profiles *Z* to minimize the difference between the observed bulk RNA-seq profiles and the abundance-weighted sums

of the expression profiles of composing cell types $X$. Namely, given bulk expression profiles $\{z_i \in Z\}$ of each mixture, and expression estimates for each cell type $\{x_j \in X\}$, we sought to identify $p_{ij} \in P$ across all mixtures $i$ and cell types $j$ to minimize the difference between the bulk RNA-seq profile and the mean expression of profiles of the composing cells (Eq. 1).

$$\underset{P}{\mathrm{argmin}}\sum_i \left(z_i - \sum_j p_{ij}x_j\right)^2 \text{ for mixture } i \text{ and cell type } j \tag{1}$$

### Transforming and deconvolving bulk RNA-seq profiles with SQUID

We present SQUID, a conversion-dampened weighted least squares strategy to transform and deconvolve bulk RNA-seq data into scnRNA-seq vector spaces. SQUID was intended to test the potential of using concurrent RNA-seq and scnRNA-seq profiling to improve deconvolution accuracy. Similar to Bisque [40], SQUID learns a transformation function of bulk RNA-seq profiles $Z$ to the concurrent pseudobulk profiles $\widehat{Z}$, where pseudobulk scnRNA-seq profiles are estimated as mean abundance across cells and samples. Then, the bulk RNA-seq expression profile of each gene $g$ with non-zero expression in both the bulk and scnRNA-seq profiles is mapped to its pseudobulk profile according to Eq. 2, where $\widehat{z}_{g,i}$ and $z_{g,i}$ are the pseudobulk and bulk profiles of gene $g$ in sample $i$, respectively. The coefficient $a_g$ and constant $b_g$ form the linear transformation for each gene $g$.

$$\underset{a,b}{\mathrm{argmin}}\sum_i \left(\widehat{z}_{g,i} - \left(a_g z_{g,i} + b_g\right)\right)^2 \tag{2}$$

This linear transformation was applied to all bulk RNA-seq profiles to transform them to scnRNA-seq space. This transformation minimizes the deviation between a sample's pseudobulk and bulk RNA-seq profiles by mapping the bulk RNA-seq expression profile of each gene to the magnitude and deviation of pseudobulk scnRNA-seq values. Equation 2 also applies when converting bulk RNA-seq profiles with no concurrent scnRNA-seq profiles. However, when testing deconvolution by SQUID on our datasets, which included concurrent bulk RNA-seq and scnRNA-seq profiles for each sample, we used a left-one-out strategy. Namely, the linear transformation was optimized using all but one sample and was then used to transform the bulk RNA-seq profile of the remaining sample. This transformed profile was then used to predict the composition of the sample with the dampened weighted least squares strategy DWLS [14]. Deconvolution performance was determined using cell counts for our cell mixtures and estimates from single-cell profiles for patient samples with concurrent bulk and scnRNA-seq profiles (gold standard). We note that cell counts are the most accurate and unbiased estimates for our cell mixtures, and single-cell estimates are our only estimates for the true composition of patient samples. Comparisons of SQUID and other deconvolution method accuracy without cross validation are given in Additional file 1: Figures S2-6.

We note that the proposed linear transformation in Eq. 2 is not unique. Indeed, Bisque proposed an alternative transformation that could be used more generally (Eq. 3). We tested this formulation and found that it performed equivalently to the formulation

given in Eq. 2. Namely, let $\overline{\widehat{z_g}}$ denote the average expression estimate of gene $g$ in pseudobulk profiles $\widehat{Z}$ and $\overline{z_g}$ the average expression of this gene in all bulk RNA-seq profiles—including the matching and other bulk RNA-seq and profiles $Z$, and let $\widehat{\sigma_g}$ and $\sigma_g$ denote their respective standard deviations. Then, the transformed profile for gene $g$ in sample $i$ ($\overrightarrow{z}_{g,i}$) is given in Eq. 3. Note that this formulation does not require RNA-seq and scnRNA-seq profiling to be concurrent.

$$\overrightarrow{z}_{g,i} = \overline{\widehat{z_g}} + \frac{\widehat{\sigma_g}}{\sigma_g}(z_{g,i} - \overline{z_g}) \tag{3}$$

Following transformation using Eqs. 2 or 3, SQUID adopts a simplification of DWLS's strategy to deconvolve transformed bulk profiles. SQUID does not require signature gene selections and instead uses all genes with nonzero expression in both the transformed bulk and scnRNA-seq profiles. The objective function is identical to the one employed by OLS (Eq. 1), however, here, the SQUID process seeks to identify $\widetilde{p}_{ij} \in \widetilde{P}$ that minimizes the discrepancy between transformed bulk RNA-seq profiles $\overrightarrow{Z}$ and the abundance-weighted sums of the expression profiles of composing cell types $X$. Consequently, following the iterative process proposed by Tsoucas et al., SQUID minimizes this dampened weighted discrepancy until convergence is reached at iteration $l$, so that $\|\widetilde{P}^{(l)} - \widetilde{P}^{(l-1)}\| \le 0.01$ [14].

### A five-step approach to determine the number of clusters in scRNA-seq data

For those datasets for which no metadata was available, we performed the following five-step iterative process.

(1) Use Monocle3 clustering (see Additional file 1: Figure S1A), which does an internal log transformation and library size normalization, to assign initial labels to all cells in each scRNA-seq dataset.
(2) Compute a mean expression profile per cluster using log-transformed and library-size normalized data from Monocle3.
(3) Compute all pairwise Pearson correlations across the mean expression profiles.
(4) Combine all non-overlapping cluster pairs with the highest correlation where Pearson correlation $r >= 0.95$ (see Additional file 1: Figure S1B).
(5) Modify the clustering information inside the metadata file (that we labeled as "phenoDataC").

### Cell mixture construction

#### *Tissue culture*

MCF7 cells were purchased from the Tissue Culture Core at Baylor College of Medicine. BT474, T47D, and THP1 cells were purchased from ATCC; Jurkat (J32) cells were a gift from Dr. Andras Heczey; hMSC cells were purchased from Lonza (PT-2501). MCF7 cells were cultured in DMEM with 10% FBS; T47D cells were cultured in RPMI with 10% FBS; BT474 cells were cultured in DMEM with 10% FBS and 15 μg/ml insulin; Thp1 and Jurkat cells were cultured in RPMI with 10% FBS and 1% L-glutamine; hMSCs were

cultured using the MSCGM BulletKit from Lonza. All cell lines were cultured with 1% penicillin–streptomycin (ThermoFisher Scientific) and maintained at 37 °C in a humidified incubator with 5% $CO_2$. All cell lines were confirmed to be free of mycoplasma contamination by DNA staining with Hoechst (ThermoFisher Scientific) or Syto 82 (ThermoFisher Scientific). DMEM, RPMI, and L-glutamine were purchased from ThermoFisher Scientific, and FBS and bovine insulin were purchased from Sigma. All cell lines were authenticated by the Cell Line Authentication Testing Division of Labcorp.

### Mixture assay

Adherent cells were harvested in a proliferative state. Cells were washed in PBS, trypsinized, collected, and resuspended in HBSS (ThermoFisher Scientific) with 10% FBS. Suspension cells were collected during the log growth phase and resuspended in HBSS with 10% FBS. All cells were maintained on ice after harvest and counted on a Countess II FL (Life Technologies). Viability was high for all cell lines, and the average of three counts was used to calculate cell concentrations. Per mixture, 16 K cells were submitted for scRNA-seq, 500 K cells were prepared for bulk RNA sequencing, and 50 K cells were prepared for flow cytometry.

### Bulk RNA isolation and sequencing

Cell pellets of approximately 500 K cells were prepared in triplicates and flash frozen at the time of the experiment. RNA was extracted using the Qiagen RNeasy Plus mini kit with a genomic DNA elimination column (74,104). RNA quality was confirmed based on RIN, and 150 bp paired-end mRNA libraries were prepared by Novogene (Sacramento, California, USA), who also sequenced libraries at a depth of 20 M reads per sample on the NovaSeq 6000 platform (Illumina).

### Single-cell RNA library preparation and sequencing

Single-cell samples were submitted to the Baylor College of Medicine Single Cell Genomics Core immediately after preparation. Per sample, 16 K cells were loaded, with an expected return of 10 K cells. Single-cell gene expression libraries were prepared according to the Chromium NextGEM Single Cell Gene Expression 3v3.1 kit (10 × Genomics). Briefly, cells, reverse transcription reagents, gel beads containing barcoded oligonucleotides, and oil were loaded on a Chromium controller (10 × Genomics) to generate single-cell GEMs (Gel Bead-In-Emulsion). Full-length cDNA was synthesized and barcoded within each GEM. Subsequently, GEMs were broken, and cDNA was pooled. Following cleanup using Dynabeads MyOne Silane Beads, cDNA was amplified by PCR. The amplified product was fragmented prior to end-repair, A-tailing, and adaptor ligation. Final libraries were generated by amplification. Sequencing of single-cell libraries was performed by the Genomics and RNA Profiling Core at Baylor College of Medicine. To reach an estimated 20 K reads per cell, samples were sequenced at a depth of 200 M reads on the NovaSeq 6000 platform (Illumina).

### Flow cytometry

Immediately after cell collection, a portion of each cell suspension was stained with Hoechst (10 µM in HBSS) or Syto 82 (5 µM in HBSS) for 10 min at 37 °C. After staining,

cells were washed and resuspended in HBSS containing 10% FBS. Stained cells were counted twice, and staining efficiency was assessed using Countess II FL. Staining efficiency was nearly 100% in all cell lines, and viability remained high. Count averages were used to calculate the number of cells added to each mixture, and 50 K cells were targeted for each flow sample. All flow samples were prepared and analyzed in triplicate. For each of the six mixtures (1–6), three flow samples (1A–1C, 2A–2C, etc.) containing one Hoechst-stained cell line (either T47D, BT474, or MCF7), one Syto 82-stained cell line (either Jurkat, THP1, or hMSC), and four unstained cell lines at identical proportions were generated. Single-stained cells from these samples represented the proportion of that cell line in the corresponding mixture. This strategy (Additional file 1: Figure S7) was developed to avoid spectral overlap and to increase our ability to accurately quantify positive cells. For each cell line, unstained and single-stained samples were used as controls to set voltages and define positive and negative gates. Flow cytometry was performed on a FACSymphony (BD Biosciences). Forward and side scatter areas were compared to select cells and exclude debris. Then, forward scatter height and area were compared to select single cells and exclude doublets. Single cells were sub-gated using positive and negative cut-offs for Hoechst (405 nm laser, BV421 channel) and Syto 82 (561 nm laser, PE channel). Gates were set independently for each cell line due to large differences in cell sizes and to maximize the number of single-stained cells. Once set, these gates were applied universally to all mixtures. Comparison of BV421 and PE areas demonstrated few double-positive cells and three distinct populations: unstained cells, BV421-positive cells, and PE-positive cells. Data were exported and analyzed using FlowJo v10.8.0 (BD Biosciences). Average flow proportions were compared to expected cell counts and showed a high correlation.

### *Pediatric AML and neuroblastoma profiling*
Paired diagnosis-relapse samples from 6 Pediatric AML patients that were enrolled in AAML1031 were profiled by CITE-seq, including scRNA-seq (Immunai), labeling RNAs with a 10 × Genomics Chromium controller and sequencing with Illumina Novaseq 600. In total, we profiled a total of 15,857 genes in 27,687 cells, with an average of 4,644 UMIs and 1432 gene features per cell (RNA only). Cells with mitochondrial gene content above 10% and fewer than 500 UMIs were excluded. AML samples were treated with RNAlater and profiled using Illumina Novaseq 600 with 25 M reads per sample. Similarly, patients in the NB1 dataset were profiled by bulk RNA-seq with Illumina Novaseq 600 with 25 M reads per sample.

## Supplementary Information

---

**Additional file 1: Supplementary Figures S1-S7.** Includes the presentation of detailed analyses and comparisons of assays and methods.

**Additional file 2: Table S1.** The composition of each of the 6 cell mixtures, based on cell counts, compared to estimates based on flow cytometry and scRNA-seq analysis.

**Additional file 3: Table S2.** Cell-type specific biomarkers for cell lines used in the 6 cell mixtures.

**Additional file 4: Table S3.** RNA-seq and scRNA-seq estimated expression profiles for cell mixtures and cell lines.

---

**Additional file 5: Table S4.** Datasets used to evaluate deconvolution accuracy. Annotations include methods for setting the gold-standard composition, quality of deconvolution, sample types, cell or nuclei counts, UMI counts, and the number of samples in each dataset.

**Additional file 6: Table S5.** Predicted abundance and clinical annotations of pediatric AML and neuroblastoma TARGET patients that were used to evaluate the outcomes-predictive value of subclone abundances in diagnostic samples.

**Additional file 7:** Review history.

### Acknowledgements
We thank Elena Denisenko and Alistair Forrest (Harry Perkins Institute of Medical Research, Australia) for providing the necessary information to match bulk and scnRNA-seq samples from the GSE141115 dataset, Alex Swarbrick, Kate Harvey, Sunny Wu, and Dan Roden (Garvan Institute of Medical Research, Australia) for providing information regarding the state of tissue for scRNAseq captures and the matching tissue that was used for bulk RNAseq ("breast" dataset), and Andras Heczey for providing Jurkat (J32) cells.

### Peer review information

### Review history
The review history is available as Additional file 7.

### Authors' contributions
P.M. and P.S. conceived and designed the study. F.A.C., M.J.N.F, X.L., T.K.M., H–S.C., and E.C. performed bioinformatics analysis. F.A.C., M.J.N.F., and X.L. implemented the code. J.E., E.K., M.J.K., D.D.B., L.V., J.M., S.R.V.H., F.W., S.J., and M.L.R contributed and profiled samples. J.E. designed assays and profiled cell mixtures. F.A.C., M.J.N.F, P.M., and P.S. wrote the manuscript and all authors contributed to the writing and provided comments.

### Funding
The results published here are in part based upon data generated by the Therapeutically Applicable Research to Generate Effective Treatments initiative. The work was supported by CPRIT awards RP180674 and RP230120, European Union's Horizon 2020 research and innovation program under grant agreement 826121, NCI awards R21CA223140 and R21CA286257, and Special Research Fund postdoctoral scholarship from Ghent University (BOF21/PDO/007). Cell mixtures were profiled by the BCM Single Cell Genomics Core, which is supported by NIH shared instrument grants S10OD018033, S10OD023469, S10OD025240, and P30EY002520.

### Availability of data and materials
*Data availability.* Pediatric AML and cell mixture scRNA-seq datasets are available at Gene Expression Omnibus (GEO) [45]. Bulk RNA-seq and scRNA-seq data for kidney [46] and breast cancer [47] samples are available at GEO. IHC profiles of bulk Brain samples are available at GEO [48]. Bulk RNA-seq and scRNA-seq data from patients in the NB2 dataset [49, 50] and bulk RNA-Seq and snRNA-Seq data from patients in the NB2 [51] datasets are available at the European Genome Phenome Archive EGA. ROSMAP brain cell profiles are available at Synapse [52]. Patient RNA-seq data used for this study, phs000467 (neuroblastoma) and phs000465 (AML), are available at TARGET's GDC portal; see Additional file 5: Table S4 for additional details on these datasets.
*Code availability.* All software and scripts used to manufacture the reported analyses, including R implementations of SQUID with and without cross validation, are available under the BSD 3-clause license at the Github [53] and Zenodo [54] repositories.

## Declarations

### Ethics approval and consent to participate
The neuroblastoma and AML studies were approved by the institutional review boards at each institution, including Ghent University and Princess Maxima Center for Pediatric Oncology (neuroblastoma), and Texas Children's Hospital Cancer Center (AML). All subjects included in this study provided written informed consent for the use of phenotype and molecular data for research. The research conforms to the principles of the Declaration of Helsinki.

### Consent for publication
Not applicable.

### Competing interests
L.V. is a shareholder of Immunai. All financial interests are unrelated to this work. The other authors declare no competing interests.

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

## 