## [**Additional file 7: ** Review history. · Genome Biology]

Review History

First round of review

Reviewer 1

Are you able to assess all statistics in the manuscript, including the appropriateness of statistical tests used? Yes, and I have assessed the statistics in my report.

Comments to author:

In their manuscript "Effective methods for bulk RNA-Seq deconvolution using scRNA-Seq transcriptomes", Cobos et al. benchmark various in silico cell type deconvolution methods on eight data sets for which matched bulk RNA-seq and single-cell RNA-seq profiles are available. Typically, benchmark studies rely on single-cell, FACS or IHC profiles as the gold standard for evaluating cell type composition. Most interesting is thus the contribution of a new benchmark data set where cell lines have been pooled in known fractions to offer an unbiased data set. Indeed, this true gold standard also gives unique insights into the bias of compositional analysis in single-cell RNA-seq. Interestingly, both in silico deconvolution of bulk RNA-seq and single-cell inferred cell type fractions correlate well with the ground truth but not so much with each other, highlighting that single-cell RNA-seq is an imperfect but workable ground truth and identifying an upper bound for the accuracy of the tested methods. In addition, the authors tested different normalization methods (on both the scRNA-seq and bulk RNA-seq data), and tested different tissue preservation methods. Finally, the authors present a new method called SQUID which is inspired by DWLS and Bisque. Like the latter, SQUID leverages matched single-cell and bulk RNA-seq data sets to learn a transformation of the bulk RNA-seq samples into the single-cell RNA-seq feature space, thus correcting for biases emerging between both technologies. The advantage is that only a few samples have to be assayed with a single-cell technique to learn the transformation, which can then be applied to a larger number of bulk RNA-seq data. The authors show that the performance of SQUID is superior to methods that can not leverage this additional information. Finally, they also present a use case where they show that a rare subclone in AML can be identified using deconvolution, demonstrating the clinical potential of this class of methods. The authors have done a lot of solid work here and present a new gold standard data set, benchmark, new method and interesting application case. The manuscript is well written and an important contribution to the field. I have the following comments:

#Major:

- A plethora of deconvolution tools have been proposed in the literature but only very few have been selected here for benchmarking. Please elaborate on the choice or consider including more alternatives (such as Autogenes etc.).
- I'm surprised that Bisque was not included in the comparison as SQUID was inspired by this technique. Is there a good explanation for this? Otherwise it should also be included in the benchmark.

- It is commendable that the authors make their source code openly available but it seems like SQUID as a method is currently rather a loose collection of scripts. Turning SQUID into a proper R package would make it considerably easier for other users to employ this tool.
- For the future you might also want to consider integrating SQUID in omnideconv which is an effort to aggregate deconvolution tools in a unified interface (<https://github.com/omnideconv/omnideconv>).
- An important confounder between single-cell RNA-seq and bulk RNA-seq is the total amount of RNA in a cell, which differs considerably by cell type. This bias is accounted for by some methods (e.g. quantiseq or EPIC) and methods like census (<https://www.nature.com/articles/nmeth.4150>) have been developed to account for this. This bias has also been evaluated here: <https://doi.org/10.1093/bioinformatics/btac499>. Some problems observed here, e.g. Figure 2H where a shift between the cell lines is visible, may be related to this. This point could be added to the discussion.
- The results presented in Figure 4C show a problem: the correlations are entirely dominated by one very abundant cell type, making the result a bit spurious. I thus have some doubts if the findings presented here with respect to different tissue preservation methods would generalize. In my opinion, the authors should discuss these results more critically.
- I really appreciate the new data set with mixtures of different cell lines but the authors should highlight limitations better. This is a rather artificial setting that does not reflect the complexity and heterogeneity found in real tissue samples. For instance, in a real bulk sample there may also be cell types that are absent or depleted in single-cell data.

Minor:

- Discussion: "However, their absolute accuracy remains unstudied" - I think you mean understudied.

Reviewer 2

Are you able to assess all statistics in the manuscript, including the appropriateness of statistical tests used? Yes, and I have assessed the statistics in my report.

Comments to author:

This study presents a systematic evaluation of cell type deconvolution algorithms by an experimentally generated scRNA-seq and bulk RNA-seq data set from a mixture of cell lines. This unique data set allows the investigators to evaluate and compare the performance of common transformation/normalisation methods and deconvolution methods. Based on their results, they further devised a method, called SQUID, that enables better deconvolution. The data set and the code are publicly available via github and other appropriate repositories. The paper is well written and generally easy to follow. I support publication of this manuscript.

Reviewer 3

Are you able to assess all statistics in the manuscript, including the appropriateness of statistical tests used? Yes, and I have assessed the statistics in my report.

Comments to author:

The study by Avila Cobos et al. carries out a benchmarking analysis of deconvolution methods and proposed a new data transformation approach (SQUID) to improve deconvolution outcomes.

The study is thorough and extensive and uses a well characterised benchmarking dataset, so I believe it will be a valuable resource in the field. I would like to see some more evidence for the effectiveness of the SQUID transformation, and there are a few key elements that would be useful to be added.

Major points.

1. I find the scnRNA-seq abbreviation which collapses scRNA-seq and snRNA-seq confusing. These are very different types of data, and one would expect them to behave differently as cell-type signature data. So a key element missing in the study is a comparison of scRNA-seq vs snRNA-seq. This is critical for tissues such as the brain where only snRNA-seq is available. Further to this, does the SQUID transformation work equally well for scRNA-seq and snRNA-seq? Since bulk RNA-seq typically profiles whole cells rather than nuclei, one would expect noticeable differences.
2. The study does not consider the effect of the RNA content of different cell types. In fact gene expression deconvolution estimates RNA proportions in the sample, rather than cell type proportions. This is equivalent to cell type proportions only if all cell types in the mixture have roughly the same size/RNA content. Is the poor outcome of deconvolution observed for certain cell types (eg. Fig3 Jurkat, THP1) explained by difference in cell size or RNA content?
3. "We studied 7 human and murine tissue datasets with concurrent RNA-Seq and scnRNA-Seq profiles." I wasn't able to find a thorough description of these datasets in methods. Please specify if these are scRNA-seq or snRNA-seq, how many samples were included in each dataset and what exactly was deconvolved. A supplementary table would be useful.

Minor points:

Some recent deconvolution benchmarking papers are missing from the references/discussion for example PMID 35292647 on tissue-specific effects in deconvolution, and PMID: 33845875.

Reviewer 1

Summary. In their manuscript "Effective methods for bulk RNA-Seq deconvolution using scRNA-Seq transcriptomes", Cobos et al. benchmark various in silico cell type deconvolution methods on eight data sets for which matched bulk RNA-seq and single-cell RNA-seq profiles are available. Typically, benchmark studies rely on single-cell, FACS or IHC profiles as the gold standard for evaluating cell type composition. Most interesting is thus the contribution of a new benchmark data set where cell lines have been pooled in known fractions to offer an unbiased data set. Indeed, this true gold standard also gives unique insights into the bias of compositional analysis in single-cell RNA-seq. Interestingly, both in silico deconvolution of bulk RNA-seq and single-cell inferred cell type fractions correlate well with the ground truth but not so much with each other, highlighting that single-cell RNA-seq is an imperfect but workable ground truth and identifying an upper bound for the accuracy of the tested methods. In addition, the authors tested different normalization methods (on both the scRNA-seq and bulk RNA-seq data), and tested different tissue preservation methods. Finally, the authors present a new method called SQUID which is inspired by DWLS and Bisque. Like the latter, SQUID leverages matched single-cell and bulk RNA-seq data sets to learn a transformation of the bulk RNA-seq samples into the single-cell RNA-seq feature space, thus correcting for biases emerging between both technologies. The advantage is that only a few samples have to be assayed with a single-cell technique to learn the transformation, which can then be applied to a larger number of bulk RNA-seq data. The authors show that the performance of SQUID is superior to methods that can not leverage this additional information. Finally, they also present a use case where they show that a rare subclone in AML can be identified using deconvolution, demonstrating the clinical potential of this class of methods. The authors have done a lot of solid work here and present a new gold standard data set, benchmark, new method and interesting application case. The manuscript is well written and an important contribution to the field.

Reply. We thank the reviewer for the thoughtful review and encouraging comments. The comments below are addressed point by point and resulted in several improvements to the presentation of the work in the revised manuscript.

Q.1.1. A plethora of deconvolution tools have been proposed in the literature but only very few have been selected here for benchmarking. Please elaborate on the choice or consider including more alternatives (such as Autogenes etc.) I'm surprised that Bisque was not included in the comparison as SQUID was inspired by this technique. Is there a good explanation for this? Otherwise it should also be included in the benchmark.

A.1.1. Our current manuscript is built on preliminary work by Avila Cobos et al. (*Nature Communications*, 2020), where we evaluated 20 deconvolution methods but focused on predictions based on pseudobulk profiles. In Avila Cobos et al. (2020), we pointed out that most deconvolution methods, including Bisque, failed to outperform OLS when tested on pseudobulk profiles, and showed that DWLS and MuSiC were the top performers on both bulk and pseudobulk deconvolution. Here, we focused on studying deconvolution of bulk RNA-Seq profiles—the most relevant data type for deconvolution applications. We have taken advantage of the increasing availability of paired bulk and single-cell RNA profiles, and have generated our own datasets for AML, NB, and cell mixtures. As pointed out by the reviewer, while we included DWLS and MuSiC in our submission, we omitted Autogenes and Bisque. Our reasons are as follows. We excluded Autogenes because its methods are similar to other methods tested; namely, Autogenes is a suite of deconvolution method implementations that are available for selection by its users. These include SVR, linear regression, and NNLS. Each of these approaches is either superseded by or is equivalent to other approaches that were tested in our manuscript, including CIBERSORT, OLS, and NNLS, respectively. We chose not to analyze

Bisque because it performed poorly in datasets tested by Avila Cobos et al. (2020) and because it doesn't allow for optimizing data normalization methods, namely, the Bisque implementation provided by Jew et al. (2020) expects its input to include raw counts from scRNA-Seq profiles, which it unavoidably converts to counts per million (CPM). Consequently, the evaluations shown in Figures 3-5 cannot be produced for Bisque. However, we were able to combine Bisque with Marker Gene optimization. The results suggested that Marker Gene optimization can lead to significant Bisque performance improvements, but SQUID, which combines the transformation procedures used by BISQUE with the optimization procedures introduced by DWLS, continued to outperform BISQUE even without Marker Gene optimization (Figure R1). This data was included in the revised Figure S6, which evaluates the deconvolution accuracy of SQUID, BISQUE when using all genes, and BISQUE after marker gene selection, and 7 other methods on 8 datasets with known or experimentally estimated cellular compositions. We added the following note in Discussions.

We note that the number of publicly available deconvolution methods is increasing rapidly, and not all methods were evaluated here. We selected methods based on their demonstrated accuracy in previous studies and based on our ability to integrate them into our benchmark implementation. Consequently, because the Jew et al. (2020) implementation of Bisque expects raw counts from scRNA-Seq profiles, we could not include it in most of our evaluation and it only appears in comparisons presented in Figure S6.

Q.1.2. It is commendable that the authors make their source code openly available but it seems like SQUID as a method is currently rather a loose collection of scripts. Turning SQUID into a proper R package would make it considerably easier for other users to employ this tool. For the future you might also want to consider integrating SQUID in omnideconv which is an effort to aggregate deconvolution tools in a unified.

A.1.2. We wholeheartedly agree with the reviewer and have produced an R package for SQUID, in addition to all the scripts used in this study. These are available on SQUID's dedicated Github

repository. Once the manuscript is published, we will also initiate a discussion with the omnideconv team to include SQUID in their ecosystem. Indeed, we have worked with omnideconv developers to improve and provide free access to academic software in the past. SQUID is freely available and should be easily integrated within the omnideconv ecosystem.

Q.1.3. An important confounder between single-cell RNA-seq and bulk RNA-seq is the total amount of RNA in a cell, which differs considerably by cell type. This bias is accounted for by some methods (e.g. quantiseq or EPIC) and multiple methods have been developed to account for this. Some problems observed here, e.g. Figure 2H where a shift between the cell lines is visible, may be related to this. This point could be added to the discussion.

A.1.3. We agree that differences in the total amount of RNA between samples may affect deconvolution estimates, but this doesn't appear to be the case in our study. In particular, shifts observed in Figure 2H are largely absent in Figure 2G. In both figures, we used OLS to deconvolve the bulk RNA-Seq profiles of mixture cells. In Figure 2G, we used bulk expression estimates of each cell line, and in 2H we used scRNA-Seq-based expression estimates. scRNA-Seq-based estimates are normalized in a cell-independent manner. Moreover, bias due to total RNA in each cell would have shifted the bulk expression profiles of each mixture and would not be resolved by differences in the profiles of individual cell lines. We believe that the cause of these shifts is the accuracy of scRNA-Seq-based expression estimates, which are cell-line dependent. This evaluation is included in Discussion. Because bulk profiles produce considerably more accurate expression estimates than pseudobulk estimates, OLS using bulk estimates (Figure 2G) is more accurate and the shifts observed in Figure 2H are likely due to systematic cell-dependent gene expression profile inaccuracies. We added a statement to this effect in Discussion, 3rd paragraph.

Q.1.4. The results presented in Figure 4C show a problem: the correlations are entirely dominated by one very abundant cell type, making the result a bit spurious. I thus have some doubts if the findings presented here with respect to different tissue preservation methods would generalize. In my opinion, the authors should discuss these results more critically.

A.1.4. We agree. The evaluation of DWLS, FARDEEP, and other methods in Figure 4C could be dominated by a single-cell type and may not generalize. However, Figure 4C analysis does clearly demonstrate poor accuracy for cryopreserved samples, even if the abundance of one cell type appears to be estimated accurately. Moreover, the correlation plots (Figure 4C right panel) also demonstrated that the predicted abundance for the less abundant cell types were heavily underestimated in cryopreserved samples compared to the gold standard composition. We amended Results section "Single-cell storage procedures impact deconvolution accuracy" to make that clear, stating the following.

We note that while the normalization choice had a relatively small impact on deconvolution accuracy for the kidney dataset, normalization may have had a strong effect on the performance of each tested method for the breast cancer dataset (Figures 4A and S2). However, our accuracy estimates were dominated by few abundant cell lines and may not generalize.

Q.1.5. I really appreciate the new data set with mixtures of different cell lines but the authors should highlight limitations better. This is a rather artificial setting that does not reflect the complexity and heterogeneity found in real tissue samples. For instance, in a real bulk sample there may also be cell types that are absent or depleted in single-cell data.

A.1.5. We completely agree with the reviewer. Our cell mixtures are a simplified model of a true solid tumor sample, for multiple reasons. We made added the following note to Discussion.

We note that our cell mixture model is a simplification of tumor samples. Namely, biological samples are often composed of more than 6 cell types, the catalog of cell types composing any 2 samples is often not equivalent, and tumor subtypes are likely to be more closely related than cell lines that are derived from unrelated patients.

Q.1.6. - Discussion: "However, their absolute accuracy remains unstudied" - I think you mean understudied.

A.1.6. We modified this discussion sentence as suggested by the reviewer.

Reviewer #2

Summary. This study presents a systematic evaluation of cell type deconvolution algorithms by an experimentally generated scRNA-seq and bulk RNA-seq data set from a mixture of cell lines. This unique data set allows the investigators to evaluate and compare the performance of common transformation/normalisation methods and deconvolution methods. Based on their results, they further devised a method, called SQUID, that enables better deconvolution. The data set and the code are publicly available via github and other appropriate repositories. The paper is well written and generally easy to follow. I support publication of this manuscript.

Reply. We thank the reviewer for the supportive comments and sincerely hope that the data and analysis presented in this manuscript will contribute to future efforts in the field.

Reviewer #3

Summary. The study by Avila Cobos et al. carries out a benchmarking analysis of deconvolution methods and proposed a new data transformation approach (SQUID) to improve deconvolution outcomes. The study is thorough and extensive and uses a well characterised benchmarking dataset, so I believe it will be a valuable resource in the field. I would like to see some more evidence for the effectiveness of the SQUID transformation, and there are a few key elements that would be useful to be added.

Reply. We thank the reviewer for the positive feedback. The manuscript includes comparisons of multiple deconvolution methods and our results suggested that SQUID outperforms all other methods using unbiased criteria. We showed that SQUID predictions are consistently more correlated and are closer to gold standard composition estimates with and without cross-validation in each of 8 independent datasets. We also showed that its improvement in accuracy has substantial value to applications of deconvolution. Namely, while all other methods failed to identify AML and NB cancer cell types that are predictive of patient outcomes, SQUID predictions were useful for identifying these cancer cells. For example, we characterized cancer cells that are present in diagnostic pediatric AML samples and expand at relapse. Only SQUID predictions showed that the abundance of these cells in bulk-profiled diagnostic samples is also predictive of outcomes. We believe that the translation of improved accuracy to application of deconvolution will be of utmost importance to personalized medicine efforts in these and other cancers. Altogether, we believe that the manuscript includes substantial evidence to support SQUID's effectiveness and practical benefit that more accurate deconvolution methods promise to the field.

Q.3.1. I find the scnRNA-seq abbreviation which collapses scRNA-seq and snRNA-seq confusing. These are very different types of data, and one would expect them to behave differently as cell-type signature data. So a key element missing in the study is a comparison of scRNA-seq vs snRNA-seq. This is critical for tissues such as the brain where only snRNA-seq is available. Further to this, does the SQUID transformation work equally well for scRNA-seq and snRNA-seq? Since bulk RNA-seq typically profiles whole cells rather than nuclei, one would expect noticeable differences.

A.3.1. We agree with the reviewer that scRNA-Seq and snRNA-Seq assays can be substantially different because the RNA populations profiled by these assays are not the same. However, multiple analyses have suggested that, on average, there is a strong association between the profiles generated by these assays, e.g., Slyper et al., *Nat Med*, 2020. Our manuscript is not focused on the differences between scRNA-Seq and snRNA-Seq assays, and we expect that expression estimates produced by them are sufficiently correlated to allow accurate and consistent cell-type mapping between cells profiled by scRNA-Seq and snRNA-Seq. In this manuscript, we focused on estimating cell abundance rather than improving expression estimates. Consequently, while it may be an imperfect solution, the scnRNA-seq abbreviation is useful in the context of this manuscript. scnRNA-Seq stands for <scRNA-Seq and snRNA-Seq>, and it was used 71 times in our manuscript because the datasets used for evaluating deconvolution methods were profiled by a mixture of scRNA-Seq and snRNA-Seq assays. When referring to scRNA-Seq or snRNA-Seq assays individually, we avoided using the scnRNA-Seq abbreviation. Consequently, the term scRNA-Seq was used 55 times in the manuscript. Finally, our tools for evaluating deconvolution methods lack the resolution to differentiate between deconvolution based on scRNA-Seq and snRNA-Seq assays, and our evaluation was based on 8 datasets, including only 2 snRNA-Seq datasets, and was not sufficiently powered to test such differences even if they could be detected. We thus insist that while making the distinction between scRNA-Seq and snRNA-Seq is biologically important, differentiating between them and studying their differences will not improve this manuscript.

Moreover, to properly compare both, one would need access to datasets where all 3 data layers—scRNA-seq, snRNA-seq, and bulk RNA-seq—are available. This is because deconvolution performance is dependent on the tissue type, as demonstrated by our study. Our results did reveal that the performance of SQUID is similar in both NB datasets (one of which was generated with scRNA-seq and the other with snRNA-seq), but additional such datasets would be required to draw any conclusions.

Q.3.2. The study does not consider the effect of the RNA content of different cell types. In fact gene expression deconvolution estimates RNA proportions in the sample, rather than cell type proportions. This is equivalent to cell type proportions only if all cell types in the mixture have roughly the same size/RNA content. Is the poor outcome of deconvolution observed for certain cell types (eg. Fig3 Jurkat, THP1) explained by difference in cell size or RNA content?

A.3.2. Indeed, the reviewer is correct to point out that Figure 3B analyses suggest that all deconvolution methods underestimate the abundances of Jurkat and THP1. However, this is not the case for deconvolution by SQUID (Figure 5C) or OLS when using bulk-derived expression profiles (Figure 2G). Our analysis suggests that bulk profiles produce more accurate expression estimates than cluster-level scRNA-Seq-based estimates. Consequently, the observed underestimates for Jurkat and THP1 are more likely to be caused by systematic scRNA-Seq technological errors that are at least partially resolved by SQUID deconvolution. If these differences were due to RNA content, then more accurate expression estimates for each cell line would not have resolved them. This point was also discussed in A.1.3 and comments were added to the revised manuscript. In general, we agree that deconvolution methods classify bulk RNA expression into cell types with no a priori information about total RNA per cell type. Addressing this potential bias should be a goal for future deconvolution efforts. However, there is no clear indication that cellular RNA content biases deconvolution results presented in this manuscript. We added a statement to this effect to Discussions, 3rd paragraph.

Q.3.3. "We studied 7 human and murine tissue datasets with concurrent RNA-Seq and scRNA-Seq profiles." I wasn't able to find a thorough description of these datasets in methods. Please specify if these are scRNA-seq or snRNA-seq, how many samples were included in each dataset and what exactly was deconvolved. A supplementary table would be useful.

Q.3.3. We agree with the reviewer and regret that this data was not easily identified in the original submission. However, our original submission included 5 supplementary tables, and Table S4 includes all the information about the datasets, including the data fields suggested by the reviewer above. This table was included in the original submission, is referred to in Introduction, and is clearly described in Supplementary Tables, and the requested information is also given in the associated Github repository.

Q.3.4. Some recent deconvolution benchmarking papers are missing from the references/discussion for example PMID 35292647 on tissue-specific effects in deconvolution, and PMID: 33845875.

Q.3.4. Following the reviewer's suggestion, we included these manuscripts in the background discussion on efforts to produce and evaluate deconvolution technologies.

Second round of review

Reviewer 1

I would like to thank the authors for answering and addressing my previous comments. I appreciate the additional work that went into setting up the R package. I have one more request: since dependencies of R packages are difficult to work with I would like to see a more easily reproducible solution to complement the existing github installation instructions, e.g. either a Renv environment, conda package or docker container that allow users to install the package without package conflicts.

Reviewer 3

The revision addresses most of my comments. The answer on scRNA-seq vs snRNA-seq was underwhelming. The authors "expect that expression estimates produced by them are sufficiently correlated to allow accurate and consistent cell-type mapping between cells profiled by scRNA-Seq and snRNA-Seq." There is published data showing that deconvolution estimates obtained using scRNA-seq and snRNA-seq are in fact sufficiently different. But I do appreciate that changing the way these data are handled in the present study would involve a major restructuring of the manuscript. If the authors are happy with the current level of ambiguity (i.e. the reader not knowing whether the signature was sc or sn in any given analysis) and don't think this impacts the overall message, then the manuscript is ready for publication from my perspective.

Authors Response

Point-by-point responses to the reviewers' comments:

As suggested by Reviewer 1, we added a Renv environment to the SQUID package. We also created a Zenodo interface for the package: doi.org/10.5281/zenodo.8127619.

We added the following note to discussions.

--We note that while we used the scnRNA-Seq abbreviation to improve readability, scRNA-Seq and snRNA-Seq assays can produce substantially different expression estimates because the RNA populations profiled by these assays are not the same. Our study was not sufficiently powered to evaluate differences between deconvolutions based on scRNA-Seq and snRNA-Seq assays, but we expect that their expression estimates are sufficiently correlated to allow for accurate and consistent cell-type mappings between scRNA-Seq and snRNA-Seq profiles.